# The Evil Twins of Chronic Pelvic Pain Syndrome: A Systematic Review and Meta-Analysis on Interstitial Cystitis/Painful Bladder Syndrome and Endometriosis

**DOI:** 10.3390/healthcare12232403

**Published:** 2024-11-29

**Authors:** Alessandra Inzoli, Marta Barba, Clarissa Costa, Valeria Carazita, Alice Cola, Martina Fantauzzi, Paolo Passoni, Serena Polizzi, Matteo Frigerio

**Affiliations:** 1Department of Medicine and Surgery, University of Milan-Bicocca, 20126 Milano, Italy; a.inzoli2@campus.unimib.it (A.I.);; 2Department of Gynecology, Fondazione IRCCS San Gerardo Dei Tintori, University of Milano Bicocca, Via Pergolesi 33, 20900 Monza, Italy

**Keywords:** endometriosis, chronic pelvic pain, interstitial cystitis/bladder pain syndrome

## Abstract

Background: Chronic pelvic pain is a debilitating condition affecting quality of life. Endometriosis is one of the leading causes of CPP, but recent studies highlighted the role of interstitial cystitis/bladder pain syndrome (IC/PBS) in causing CPP. Only some studies addressed the coexistence of these two conditions, which seems more frequent than what is supposed, leading to diagnostic delays and unnecessary surgeries. This systematic review aimed to evaluate the estimate of the prevalence of the comorbidity of endometriosis and IC/PBS. Methods: We performed a systematic review of the literature indexed on PubMed, Scopus, ISI Web of Science, and Cochrane using a combination of keywords and text words represented by “painful bladder syndrome”, “endometriosis”, “interstitial cystitis”, and “bladder pain syndrome”. We performed a meta-analysis of the results. Results: The meta-analysis shows that the coexistence of endometriosis and IC/PBS in women with CPP ranged from 15.5% to 78.3%, which is higher than the prevalence of IC/PBS in the general population. Conclusions: Prevalence data about the coexistence of endometriosis and IC/PBS are highly heterogeneous, probably due to the paucity of available data. However, in cases of endometriosis unresponsive to treatment, other reasons for CPP (such as IC/PBS) need to be ruled out.

## 1. Introduction

Chronic pelvic pain (CPP) is a debilitating condition that affects between 6 and 25% of women worldwide, depending on the definition used [1,2,3,4,5]. CPP is linked to poor health-related quality of life and decreased work efficiency [6]. Indeed, a recent systematic review estimated that the yearly cost of CPP is USD 2.8 billion [7]. CPP accounts for nearly 10% of gynecology consultations, 12% of hysterectomies and more than 40% of diagnostic laparoscopies [8,9]. However, the WHO defined it as “a neglected reproductive health morbidity” since, despite its significance, healthcare planning ignored it and failed in resource allocation due to a lack of primary epidemiological data [6]. The American College of Obstetricians and Gynecologists (ACOG) and the ReVITAlize initiative define CPP as the presence of non-cyclic pelvic pain that lasts for 6 months or longer, unrelated to pregnancy, that can be exacerbated by sexual intercourse or menstrual cycles and is often associated with negative cognitive, behavioral, sexual and emotional consequences [10,11]. In addition to this, studies reveal a frequent delay in the diagnosis with up to 50% of women without one even after many years of follow-up [2,12,13].

The pain is related to the pelvis, and both patients and clinicians localize the pain as perceived there. The experience of pain is the result of activities within the central nervous system (CNS) [14]; therefore, women with CPP have changes in brain morphology or function similar to the ones with other chronic pain conditions [15]. These changes activate specific brain regions and the hypothalamic-pituitary-adnexal axis linked to increased psychological distress [16].

Central sensitization is important for the perpetuation of chronic pain syndromes since it explains allodynia (feeling of pain in response to innocuous stimuli) and hyperalgesia (feeling a heightened response to painful stimuli) [16].

There are differential diagnoses for CPP (such as adenomyosis, endometriosis, intrauterine pathology, diverticulitis, and inflammatory bowel disease [17]), and ACOG recommends organizing the possibilities into visceral, neuromusculoskeletal and psychosocial contributors (Table 1); however, it is pivotal to maintain awareness of the multifactorial etiology, which requires an interdisciplinary model of care [18].

Endometriosis is a chronic disease that causes CPP in up to 80% of cases [19,20]. Endometriosis symptoms are commonly dysmenorrhea, dyspareunia and perimenstrual lower abdominal pain; however, patients with endometriosis may refer to dyschezia, dysuria and even frequent urination [20,21]. Indeed, symptoms have little correlation to the extent of endometriosis; therefore, in a woman with CPP, even if endometriosis is found it is always mandatory to consider other causes. After endometriosis is suspected, the confirmation is the laparoscopic visualization with positive biopsy [20]; however, the latest guidelines of the European Society for Reproductive Medicine (EHSRE) [22] also highlighted the role of ultrasound evaluation and empirical treatment as the first step in the management of patients with endometriosis.

Interstitial cystitis/bladder pain syndrome (IC/BPS) is a chronic condition characterized by an unpleasant sensation (pain, pressure or discomfort) perceived to be related to the urinary bladder, associated with lower urinary tract symptoms of more than six weeks duration in the absence of any other identifiable pathology (such as urinary tract infections, bladder carcinoma or cystitis) [23].

It was thought that this was a rare condition; however, now it is known that the prevalence is between 2.7% and 6.5% [24] of all women, even though a recent epidemiology survey [25] estimated the prevalence of IC/PBS in the general population at 0.87%.

According to the results of the Interstitial Cystitis Data Base (ICBD) Study sponsored by the National Institute of Diabetes and Digestive and Kidney Disease (NIDDK), 93.6% of classic interstitial cystitis patients report various degrees of pain [26], mainly located in the lower abdomen, in the lower back, in the vaginal area and in the rectum. The other most frequent symptom is urgency (80.4% of patients).

According to the European Association of Urology [26] and the European Society for the Study of Interstitial Cystitis (ESSIC) [27] the diagnosis of IC/PBS is based on clinical symptoms. Since it is a clinical diagnosis, it is important to exclude other diseases as the cause of the pain, such as carcinoma of the bladder, infections, radiation, and chemotherapy. Cystoscopy with hydrodistension and biopsies can document positive signs of PBS, making the diagnosis more probable, especially in patients in which the presence of Hunner’s lesions is more frequent (patients older than 50 years old) or for whom first-line treatment has failed [28]. According to cystoscopy findings, sub-phenotypes of IC/PBS were found. Patients who are Hunner’s lesions-negative and have non-low anesthetic bladder capacity (>400 mL) have a non-bladder centric phenotype that is more commonly associated with systemic pain disorders [29].

The purpose of this systematic review was to estimate the prevalence of the coexistence of endometriosis and IC/PBS in women with CPP. Previous papers have suggested a strong relationship between IC/PBS and endometriosis, also named the evil twins. However, only a little data are available, mainly in small case series. Consequently, this systematic review aimed to estimate the prevalence of the coexistence of endometriosis and IC/PBS in women with CPP.

## 2. Materials and Methods

The study protocol was registered in PROSPERO (CRD42024536875). The systematic review was carried out according to the Preferred Reporting Items for Systematic Review and Meta-Analysis (PRISMA). We performed a systematic search of literature indexed on PubMed, Scopus, ISI Web of Science, Cochrane (from the start to 4th of March 2024—start date of search), using EndNote x8 (Clarivate Analytics. In three separate searches, we used a combination of keywords and text words represented by “painful bladder syndrome” AND “endometriosis”; “interstitial cystitis” AND “endometriosis”; and “bladder pain syndrome” AND “endometriosis”. A complete search strategy is provided in Figure 1. Four reviewers (AI, VC, MF, MB) independently screened titles and abstracts of the records that were retrieved through the database searches. No article-type restrictions were applied. We only considered articles in the English language. We also performed a manual search to include additional relevant articles, using the reference lists of key articles. Full texts of records recommended by at least one reviewer were screened independently by the same two reviewers and assessed for inclusion in the systematic review. Disagreements between reviewers were solved by consensus. Data selection and extraction were conducted in accordance with PICOS (Population, Intervention, Comparison, Outcome, Study type) using a piloted form specifically designed for capturing information on the study and characteristics. Data were extracted independently by two authors to ensure accuracy and consistency. Selected studies evaluating the prevalence of IC/PBS and endometriosis in women with CPP were selected, and a meta-analysis was performed using the current version of Review Manager (RevMan) software from the Cochrane Collaboration. Means and standard deviations were used to calculate standardized mean differences (SMD) and their 95% confidence intervals (CI). We considered a *p*-value less than 0.05 to be significant. Heterogeneity was assessed using the I-squared statistic, with an I-squared value exceeding 50% indicative of substantial heterogeneity. The PRISMA 2020 checklist of this study is shown in Appendix A. We excluded items unrelated to the research question based on title, abstract or full-text reading, articles not in English, and partial articles. Due to the epidemiological nature of this systematic review, which mostly focuses on the diagnosis of this coexistence, for the quantitative analysis, we included cohort, prospective and retrospective studies.

## 3. Results

### 3.1. Study Assessment

The electronic database search provided a total of 2136 results. After duplicate exclusion, there were 705 citations left. Of them, 673 were not relevant to the review based on title and abstract screening, and 5 studies were excluded since they were not full articles. Twenty-seven studies were considered for full-text assessment, of which 13 were excluded for the following reasons: 2 book chapters and 11 papers were excluded for being in languages other than English. No paper was added through reference list searching. Overall, 14 studies met the inclusion criteria and were incorporated into the review process. The papers included mostly cohort studies (including retrospective and prospective ones); they were all published after 2002 with the most recent in 2016 [31,32,33,34,35,36,37,38,39,40,41,42,43,44].

### 3.2. Main Findings

This review includes a total of 747 patients with endometriosis and IC/PBS, and the characteristics of the studies are listed in Table 2.

#### 3.2.1. Prevalence

The prevalence of the coexistence between IC/PBS and endometriosis depended on the reference population.

#### 3.2.2. The Prevalence of IC/PBS in Women with Endometriosis

Wu et al. [44] published a population-based study of patients with endometriosis and random controls. During a follow-up of three years, IC/BPS was diagnosed in 0.20% of patients with endometriosis and 0.05% of patients without endometriosis. The hazard risk (HR) for the development of IC/PBS in patients with endometriosis was 3.74 after adjusting for comorbid associations (diabetes, hypertension, coronary heart disease, obesity, hyperlipidemia, chronic pelvic pain, irritable bowel syndrome, fibromyalgia, chronic fatigue syndrome, depression, panic disorder, migraine, sicca syndrome, allergy, asthma and overactive bladder). Although a small number of patients developed IC/PBS during follow-up (30 subjects with a sample size of 36.764 patients), this study suggests that endometriosis is associated with BPS/IC. Indeed, a possible explanation for this small number is the relatively short follow-up time and the identification of the diagnoses through ICD-9-CM codes, which may exclude patients not classified correctly.

Smorgick et al. [42] retrospectively evaluated young women with a surgical diagnosis of endometriosis by the age of 21 years for the presence of other comorbid pain syndromes. Most patients had stage I or II endometriosis (84%). IC/PBS was found in 16% of patients with endometriosis. As in the previously cited study, the follow-up period was short (25 months on average), and this can explain the low number of patients with IC/PBS who were identified.

#### 3.2.3. The Prevalence of IC/PBS and Endometriosis in Women with CPP

Most studies, however, focused on the diagnosis of IC/PBS in women with CPP since it has been underestimated for many years. In the studies included in this review, data about the coexistence of IC/PBS and endometriosis in women with CPP were present.

Chung et al. [33] were among the first to verify the association between IC/PBS and endometriosis. They performed a retrospective study in 2002, including patients with CPP who underwent laparoscopy, cystoscopy and hydrodistension. Interestingly, of the 60 patients included, 58 were diagnosed with IC/BPS, and 56 had received a diagnosis of endometriosis; of them, 48 had biopsy-confirmed lesions at laparoscopy, while 8 had negative laparoscopy. In the group of patients with IC/BPS, 47 (81%) had a history of biopsy-proven endometriosis, while 7 had a history of endometriosis with negative biopsies at diagnostic laparoscopy. In the group of patients with a history of endometriosis, 54 (96.6%) had a diagnosis of IC made by cystoscopy or hydrodistension. This study highlights the high prevalence of both these conditions in patients suffering from CPP. In addition, the presence of IC/PBS in women with a diagnosis of endometriosis not confirmed by diagnostic laparoscopy may mean that in those patients the reason for CPP was IC/PBS and not endometriosis. Indeed, even if CPP may be a symptom of IC/PBS, in this study, cystoscopy revealed findings suggestive of IC/PBS even in patients asymptomatic for urinary symptoms (13, 22.5% of patients with IC/PBS). Therefore, the authors suggest that cystoscopy should be routinely performed to avoid missing the diagnosis of IC/PBS in CPP patients.

In 2005, the same authors [32] prospectively evaluated the presence of both endometriosis and IC/PBS in patients with CPP through diagnostic laparoscopy, cystoscopy and the potassium sensitivity test (PTS). Of the 178 patients with CPP evaluated, 115 (65%) had both IC and endometriosis. However, they included patients with CPP and bladder base/anterior vaginal wall and uterine tenderness with or without voiding symptoms. Therefore, the prevalence of endometriosis and IC/PBS in this study could be overestimated more than that of patients with CPP without these signs.

In a prospective study, Clemons et al. [34] evaluated the presence of IC/PBS in patients scheduled for a diagnostic laparoscopy for CPP. They made the diagnosis of IC/PBS with a combination of urgency, frequency or nocturia and positive cystoscopic findings. In their sample size of 45 women, 17 (38%) had IC/PBS, and 21 (48%) had endometriosis; however, seven patients with endometriosis were diagnosed with IC/PBS (15.5% of women with CPP).

In this study, the presence of IC/PBS was not associated with laparoscopic findings, which may highlight the need to perform cystoscopy regardless of the presence of endometriosis or adhesion. However, according to all the new diagnostic criteria for IC/PBS, it is not mandatory to perform cystoscopy to make the diagnosis. Secondly, all women with endometriosis had stage I or II (only one woman had bladder endometriosis), according to the American Fertility Society classification. Therefore, the coexistence of IC/PBS and endometriosis is also found in the early stages of the disease.

Also, Cheng et al. [31] evaluated the prevalence of IC/PBS in patients with CPP. The percentage of IC/PBS was different depending on the diagnostic criteria that were used (IC in the presence of CPP with at least one urinary symptom and the presence of glomerulations at cystoscopy; PBS as per the ESSIC [27] definition, which is a clinical one without the confirmation by cystoscopy). In their population, 50% of those with endometriosis had PBS and 60% of women with PBS had endometriosis. Interestingly, in this study, a high prevalence of urinary symptoms was found in women with dysmenorrhea (94%).

Rackow et al. [40] focused on young women ages 13 to 25 with CPP. They performed diagnostic laparoscopy and cystoscopy in 28 patients referred for CPP.

Eleven patients (39%) were diagnosed with IC and 18 (64%) with endometriosis. The coexistence of these two conditions was found in seven (25%) cases. In these patients, there was no association between some urinary symptoms (urgency or nocturia) and IC, and this is probably linked to the natural history of the disease and the age of the patients included. In this study, the routine evaluation of symptoms did not differ between IC and endometriosis, suggesting the need to evaluate both the pelvis and bladder constantly; however, a possible explanation for this is that the authors did not use validated questionnaires since the retrospective nature of this study.

Paulson et al. performed two studies, and since the latter was prospective and had similar inclusion criteria [38,39], we cannot exclude that some patients were included in both of them. Therefore, we will focus on the results from the last one.

They evaluated 284 patients with CPP who underwent a cystoscopy or laparoscopy. Of them, 172 (61%) had both endometriosis and interstitial cystitis.

Finally, Stanford et al. [41] screened women with CPP through diagnostic laparoscopy and PST. Of the 64 women included, 48 (69%) had positive PST, 18 (28%) biopsy-proven endometriosis and 41 (64%) adhesions. In total, 42% (27 patients) had positive PST and a diagnosis of endometriosis or adhesions or both. The authors did not specify the prevalence of a positive PST in women with only endometriosis (excluding the ones with adhesions).

After excluding studies where there was no distinction between patients with endometriosis and adhesions [41] and where the absence of both of them was not evaluated [32,33], we performed a meta-analysis of women with CPP with or without endometriosis considering the presence of IC/PBS as the event, including only studies where women with CPP were evaluated with laparoscopy and concurrent cystoscopy [31,34,39,40] (Figure 2).

#### 3.2.4. The Prevalence of Endometriosis in Women with IC/PBS

Overholt et al. [37] evaluated, as the reference population, women with non-bladder centric IC/PBS through the use of a registry. The patients were divided into women with a known history of endometriosis and women without. Of all women with IC/PBS, 19% had co-occurring endometriosis. Compared to patients without the coexistence of endometriosis, patients with both of them had a higher prevalence of irritable bowel syndrome (IBS), CPP, fibromyalgia and vulvodynia.

Also, Warren et al. [43], through a case-control study, found that the prevalence of endometriosis was higher in women with IC/PBS than controls (20% vs. 6%) with an odds ratio of 3.6. This, again, was also true for migraine, IBS and fibromyalgia.

Both these studies suggest that some patients with IC/PBS have a more systemic syndrome not confined to the bladder.

#### 3.2.5. Burden of Coexistence Between Endometriosis and IC/PBS

As we have seen, the coexistence between endometriosis and IC/PBS is frequent regardless of the population studied. For this reason, over the years, it is possible that many useless surgeries have been performed in patients with CPP for the differential diagnosis of CPP and for the management of endometriosis unresponsive to medical treatment. Ingber et al. [35] pointed this out in their study, which evaluated the rate of pelvis surgeries patients with IC/PBS perform.

They performed a retrospective cross-sectional case-control study including 406 women with established diagnoses of IC/PBS from clinical databases and 5000 randomly matched controls. This was part of a larger study evaluating risk factors, natural history and comorbidities of IC [45]. Patients with IC/PBS more frequently reported all types of pelvic surgery included than control. In particular, they more often performed hysterectomy, bladder suspensions, laparoscopic pelvic surgeries, dilatation and curettage (D&C). Interestingly, some surgeries were performed before or the same years as the diagnosis of IC/PBS was made (68.4% of all hysterectomies before and 10.5% the same years; 25% of all bladder suspensions before and 39.3% the same year), while some were mostly made after the diagnosis of IC/PBS (60% of cystocele repairs and 66% of rectocele repairs). In addition, in this study, women with IC/PBS were more commonly diagnosed with endometriosis and fibroids than controls.

Lentz et al. [36] focused retrospectively on women with intractable IC who were referred to a tertiary urology center. Twenty-three of them had symptoms that fluctuated during menstrual cycles with premenstrual exacerbation of pain. Of these, 18 underwent diagnostic laparoscopy, and in 10 of them, a diagnosis of endometriosis was made. Fifteen women (out of the 18 who underwent surgery) were treated with hormonal therapy, 9 with a GnRH analog and 6 with cyclic oral contraceptive pills (OCPs). Thirteen (87%) of the women who were treated with hormonal therapy had an improvement in symptoms; this accounts for all women with a coexistence of endometriosis and IC/PBS except for two women.

This study underlines the role of excluding other pain disorders in women with intractable IC. This is pivotal since we know that endometriosis is a chronic condition that can progress over time. Therefore, a punctual diagnosis is mandatory to prevent higher stages of disease.

#### 3.2.6. Clinical Evaluation and Diagnosis

Paulson et al. [39] focused on the role of anterior vaginal wall tenderness (AVWT) as a diagnostic marker for the coexistence of endometriosis and IC/PBS. AVWT was present in 96% of patients with only IC and in 39% of patients with only endometriosis. However, when evaluated in patients with both of them, the percentage rose to 94%. This underlines the use of prior physical examination to help rule out the diagnosis of endometriosis with IC/PBS.

## 4. Discussion

Patients with CPP can face difficulties in the evaluation and management of their pain. For years, endometriosis has been considered the leading cause of CPP; however, nowadays we know that not all women with endometriosis have pain and that, even in the presence of endometriosis, gynecologists need to have a high suspicion for other diseases [19,20,46]. In recent years, the possible coexistence of endometriosis and IC/PBS raised awareness, thus leading to different studies. This comes from the increasingly evident need to find an answer to chronic pelvic pain for women who live with psychological distress and who underwent surgeries without beneficial effects. Indeed, few studies [34,46] demonstrated that women with IC/PBS undergo more frequent pelvic surgery than healthy controls. Similarly, Lentz et al. [36] successfully treated women with IC/PBS who were refractory to all standard therapy for IC/PBS with hormonal therapy, which is commonly prescribed for endometriosis. Notably, in this study, the mean duration of symptoms was 9.5 years (range 1–26).

This justifies the term “evil twin syndrome”, which was coined by Chung et al. [33] to refer to the coexistence of endometriosis and IC/PBS (Figure 3).

Studies from the literature suggest that in women with CPP, the coexistence of endometriosis and IC/PBS is frequent. Table 3 summarizes the results from the studies; the prevalence ranges between 15.5 and 78.3%. Indeed, the difference between the prevalence of endometriosis and IC/PBS alone in the studies is heterogeneous, thus justifying the wide range of the prevalence of the coexistence. This is particularly true for the IC/PBS diagnosis, which relies on different diagnostic criteria through studies. Indeed, only the study by Cheng et al. [31] also considered the ESSIC criteria for the diagnosis. Considering that the prevalence of endometriosis in women with CPP is around 80% [19,20], studies with much lower prevalence probably had minimal selection bias. Unfortunately, there is no precise data about the prevalence of IC/PBS in women with CPP. Therefore, it is not easy to make the same conclusions.

The results of the meta-analysis show that the prevalence of IC/PBS in women with endometriosis is higher than in women without endometriosis. However, the result lacks statistical significance (OR 0.82; IC 0.54–1.26). This may be linked to the heterogeneity of the prevalence of the studies included in the meta-analysis, which ranges from 15.5 to 66%.

Clemons et al. [34] performed a prospective study and excluded women with a previous diagnosis of IC/PBS, thus maybe leading to a lower prevalence in women with CPP (37.8%) than in other studies. However, looking at the prevalence of endometriosis, it is lower than the prevalence reported in women with CPP. In addition, looking at women with endometriosis, up to 30% had IC/PBS. We decided to exclude women from the meta-analysis with the diagnosis of “adhesions” instead of “endometriosis”. In this study, there were 10, and of them, 4 (40%) had received the diagnosis of IC/PBS. We cannot exclude that some women with the diagnosis of “adhesions” had concurrent endometriosis, since fibrosis may correspond to the change in peritoneal lesions occurring over time [47,48]. If we also consider women with adhesions, the prevalence of the “evil twins” syndrome would be 24.4%. Finally, the relatively low sample size of this study may justify this.

Conversely, Paulson et al. [39] report a high prevalence (66%). Actually, in their cohort of women with CPP, the prevalence of endometriosis and IC/PBS itself were remarkably high (78% and 81%), thus leading to a high prevalence of these two conditions (66%). The study’s primary outcome was not to evaluate the prevalence of the coexistence. Indeed, they included patients who had already undergone laparoscopy and cystoscopy; therefore, this higher prevalence than one of the other studies may be linked to selection bias. The study of Rackow et al. [40] shows a prevalence of 30%. However, the main difference in this study is that they included only young women under 25 years old. Therefore, even if it highlights that even young patients may suffer from both conditions, the population included differs significantly from the ones from the other studies. This is an important bias when considering this study, above all because both endometriosis and IC/PBS are more commonly diagnosed in women > 25 years of age. However, as we can see from Table 2, in most studies included in this review, patients younger than 25 years old were included.

Finally, Cheng et al. [31] reported a percentage of 30% for the coexistence of these two conditions in women with CPP. The most compelling point of this study is that they proposed using ESSIC criteria for the diagnosis of IC/PBS; according to that, the estimated prevalence of the coexistence is 30%. Indeed, they included women with visually proven endometriosis even if the histological biopsy in 85 (94% of all endometriosis women) and 8 (9%) of the histological reports did not confirm the laparoscopic findings.

Due to the exclusion diagnosis of IC/PBS, the fact that many patients are asymptomatic, and IC/PBS may overlap with other painful syndromes, the true prevalence of IC/PBS in the general population is difficult to determine [23]. A recent epidemiology survey [25] estimated the prevalence of IC/PBS in the general population at 0.87%. This is lower than what we reported for patients with coexistent endometriosis (even without statistical significance) and highlights the shared pathway for the genesis of chronic pelvic pain syndromes. One of the central limits in evaluating the prevalence of this coexistence is the need for more solid data about the prevalence of endometriosis and IC/PBS taken alone. Since we lack strict guidelines or diagnostic procedures for IC/PBS, the prevalence in women with CPP varies significantly across countries, and many studies probably underestimate it [47].

In the same way, the exact prevalence of endometriosis is not known; however, nowadays, we see that it is much more common than what was thought before, even if it remains underdiagnosed and with an essential diagnostic delay [48]. In addition, the laparoscopic visualization of endometriosis may be influenced by the expertise of the surgeon, and the presence of active lesions can be hidden by long-term hormonal therapy. Finally, the difference between studies may be linked to the diagnostic criteria they used for the diagnosis of IC/PBS and above all to the selection bias. It is reasonable that a urogynecological center selected more women with IC/PBS than a center specialized in endometriosis. This is supported by the fact that the prevalences are similar when studies are performed by the same authors. Unfortunately, few studies evaluated the diagnosis of IC/PBS after this term was proposed by ESSIC in 2008 [27].

Hysterectomy is the most common non-obstetric surgical procedure among women, even if its prevalence depends on the age of women (2.8% of women ages 18–44, 22.1% of women ages 45–64) [49,50]. As we reported, CPP is the reason for hysterectomy in up to 12% of women [8,9]. However, nearly 20–25% [51,52] of patients will undergo surgery without relief in pain; therefore, hysterectomy should be considered only after the exclusion of other diseases (gynecological or otherwise). As for endometriosis, IC can also have a significant delay in the diagnosis. Indeed, Driscoll et al. [53] found that patients are symptomatic for IC for a median of 5 years before being diagnosed; patients present more frequently with just one symptom (in 89% of cases) and progressively develop the full spectrum of symptoms. This is consistent with the higher rate of surgery in women with IC/PBS. In addition, in women with persistent pain after hysterectomy, IC/PBS is diagnosed at high rates and can be resolved with the correct treatment [54]. Even if the causative role of surgery in the development of IC/PBS cannot be excluded at all since pelvic surgeries can harm the physiology of the bladder [35], it is more plausible that many women with endometriosis refractory to medical treatment who are counseled about surgical treatment with hysterectomy, already have IC/PBS. This means that even if the diagnosis of IC/PBS is of exclusion, the presence of endometriosis does not exclude the presence of IC/PBS. However, the fact that a hysterectomy may not cure the symptoms, or the disease is also probably dependent on the radicality of the surgery or the centralization of pain, and European guidelines suggest counseling women that hysterectomy may not be a definitive treatment [22].

Women with symptoms of IC/PBS may also report the presence of dysmenorrhea or dyspareunia less since they do not link the symptoms together; from this point of view, the use of a validated questionnaire may play a role in screening for IC/PBS. Concerning this, a multidisciplinary approach may help in overcoming the diagnostic limits of these two conditions and in treating both of them correctly.

Furthermore, it is essential to remember the pivotal role of the central sensitization of these women. Levesque et al. [55] developed a clinical tool for identifying women with central sensitization, finding women more willing to be unresponsive to treatment and less linked to an organic dysfunction. This tool was used successfully in the study by Cardaillac et al. [56], where, again, the presence of central sensitization was linked to the coexistence of different pain syndromes, confirming the promising role of this clinical tool. Presumably, the diagnostic delay may also improve the chance of central sensitization, urging a multidisciplinary approach, including that of a pain specialist.

Even if the results of this meta-analysis lack statistical significance, they help address and improve the research on this topic. We strongly encourage the multidisciplinary approach to women with CPP since the chameleonic aspect of it, the multidisciplinarity, is the only way to manage all the aspects of CPP, preventing the central sensitization, the “chronicity” and the detrimental effects on the life of women. It also means that even if a formal diagnosis is made, if the patient is unresponsive or only partially responsive to treatment, it is mandatory to think that there may be coexistence with other painful diseases that need evaluation and management. This is particularly true for women with endometriosis, for whom the definitive surgery is the final stage of treatment when medical treatment fails.

The first-line approach for endometriosis is medical therapy with a combined contraceptive pill or progestin therapy [22]. When treatment fails, the second-line approach is surgery. The radicality of surgery depends on the desire for childbearing; when a woman has ended her childbearing desire, the choice is for a hysterectomy. The recurrence of endometriosis after hysterectomy is extremely low and depends on the radicality of the surgery and on the concomitant bilateral ovariectomy [57,58].

First-line treatment of IC/PBS is with oral drugs (pentosan polysulfate sodium, hydroxyzine, amitriptyline, pregabalin). Several intravesical therapies can be used, such as intravesical hyaluronic acid/chondroitin sulfate, in combination or not with oral drugs [59].

When there is a centralization of pain, other treatments can be proposed. Perineal and pelvic pain share some aspects with the complex regional pain syndrome. Therefore, the literature suggests they follow a “CRPS model”. The “CRPS model” is probably helpful since sympathetic blocks [60], drugs common to these diseases (antidepressants, anti-epileptics), or well-conducted physiotherapy are effective. Provocatively, this model also encourages the adoption of preventive strategies, such as avoiding surgical procedures, which may accentuate painful symptoms, just as for CRPS. The standard pain treatments available include maintenance therapies such as tricyclic antidepressants and antihyperalgesic agents, as well as more targeted approaches for sensitization, including ketamine. Other options are transcutaneous electrical nerve stimulation techniques, specialized physiotherapy, and mind-body therapies. These treatments can be recommended to patients by specialists such as urologists, gastroenterologists, and gynecologists, emphasizing the importance of a multidisciplinary approach to management in this context [18].

The main strength of this study is the comprehensive search strategy that includes significant databases. The main limitation is the variability of the inclusion criteria used in different studies and the bias most likely linked to the kind of referral (urogynecological center, pelvic pain/endometriosis center).

## 5. Conclusions

The diagnosis of both endometriosis and IC/PBS requires specific expertise, so women should be referred to a center with a multidisciplinary approach. Additionally, due to the consistent burden of IC/PBS in women with endometriosis, it should be considered in cases of endometriosis pain unresponsive to treatment.

In 2013, Tirparlu et al. [61] published a similar systematic review; however, after that year, just two other studies were published according to our research strategy. This, despite CPP being one of the most intriguing aspects of gynecological pathology, has a detrimental effect on women’s quality of life. There is a call to action for more studies focusing on this topic to improve our knowledge about women’s care.

## Figures and Tables

**Figure 1 healthcare-12-02403-f001:**
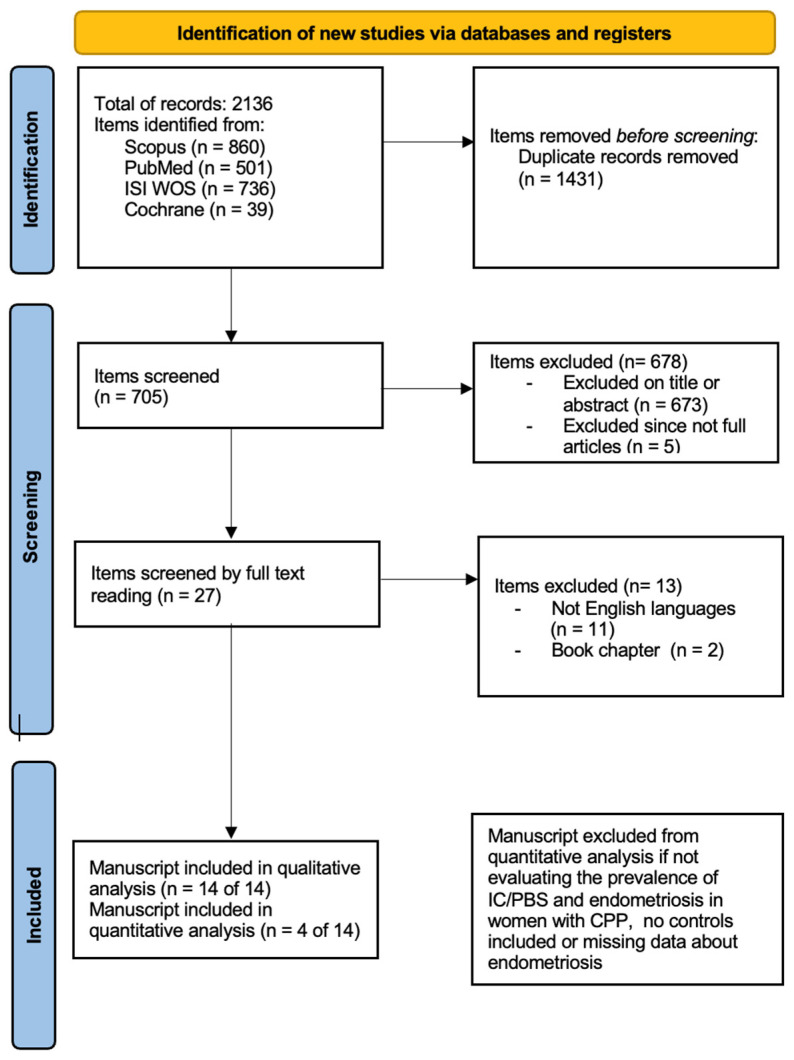
Search strategy according to PRISMA 2020 [30].

**Figure 2 healthcare-12-02403-f002:**
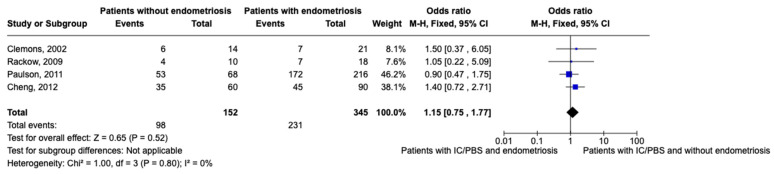
Meta-analysis of included studies (events means the presence of IC/PBS) [31,34,39,40].

**Figure 3 healthcare-12-02403-f003:**
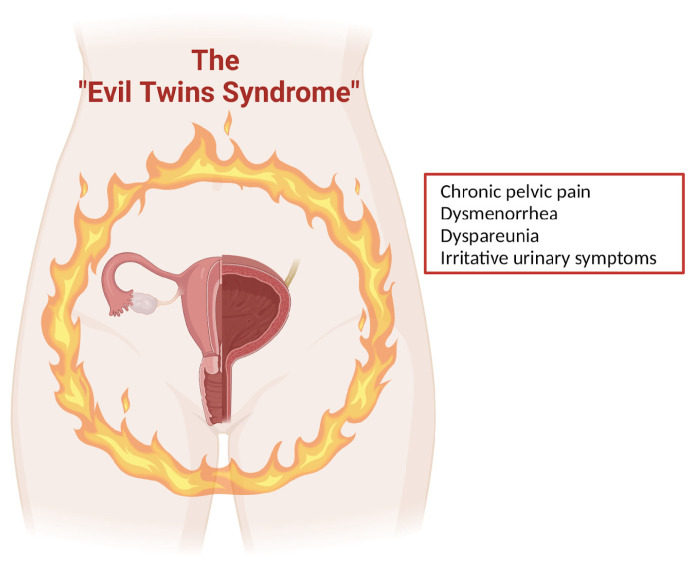
The “Evil Twins Syndrome” (created with BioRender.com).

**Table 1 healthcare-12-02403-t001:** Differential diagnosis for chronic pelvic pain. Adapted from ACOG [18].

**Visceral**	*Gynecologic*	Adenomyosis
		Adnexal mass
		Endometriosis
		Ovarian remnant syndrome
		Pelvic adhesions
		Vestibulitis
		Vulvodynia
	*Gastrointestinal*	Celiac disease
		Colorectal cancer and cancer therapy
		Diverticular colitis
		Inflammatory bowel disease
		Irritable bowel syndrome
	*Urologic*	Bladder cancer and cancer therapy
		Chronic or complicated urinary tract infection
		Interstitial cystitis
		Painful bladder syndrome
**Neuromusculoskeletal**	*Fibromyalgia*	
	*Myofascial syndromes*	Coccydynia
		Musculus levator ani syndrome
	*Postural syndromes*	
	*Abdominal wall syndromes*	Muscular injury
		Trigger point
	*Neurologic*	Abdominal epilepsy
		Abdominal migraine
		Neuralgia
		Neuropathic pain
**Psychosocial**	*Abuse*	Physical, emotional, sexual
	*Depressive disorders*	Major depressive disorders
		Persistent depressive disorders (dysthymia)
		Substance-induced or medication-induced anxiety disorder
**Somatic symptom disorders**	Somatic symptom disorders with pain features or somatic characteristics	
**Substance use disorder**	Substance abuse or dependence	

**Table 2 healthcare-12-02403-t002:** Characteristics of the studies included.

First Author	Year	Country	Study Design	N	Reference Population	Age	Diagnostic Criteria	Exclusion Criteria	Cases	FUP
Cheng [31]	2012	AU	C	150	CPP	18–50	C, BH, LPS (visually proven), patients’ symptoms, according to ESSIC criteria	Positive urine cultures, non-English speaking	27	-
Chung [32]	2005	USA	C	178	CPP	18–60	C, BH, LPS (biopsy proven), PST	Positive urinary cytology and urine and genital cultures	115	-
Chung [33]	2002	USA	C	60	CPP	19–62	C, BH, LPS (biopsy proven)	Positive urinary cytology and urine and genital cultures	47	-
Clemons [34]	2002	USA	C	431	CPP	20–53	C, BH, urinary symptoms, LPS (biopsy and/or visual proven)	Prior diagnosis of IC, urinary tract infections in the past 4 weeks, vaginal infection, bladder or gynecological cancer, urinary tract calculi, urethral diverticulum, cystitis due to chemotherapy, radiation or tuberculosis	7	-
Ingber [35]	2008	USA	C	5406	IC/PBS and controls	50.6	C, BH, for endometriosis patients self-report	-	55	-
Lentzl [36]	2002	USA	C	46	Intractable IC/PBS	23–48	C, BH, LPS (biopsy proven)	Pregnant status, history of endometriosis, previous treatment with gonadotropin-releasing hormone analog	10	-
Overholt [37]	2020	USA	C	431	IC/PBS	18–80	Medical records	Urogenital cancer, urethral diverticulum, neurologic disease, cyclophosphamide use, radiation cystitis, bladder tuberculosis, current urethral catheter placement, active bladder infection	82	-
Paulson [38]	2007	USA	C	123	CPP	-	LPS (biopsy or visual proven), C, BH	Menopausal status, musculoskeletal, gastrointestinal or other non-gynecological and urological conditions	107	-
Paulson [39]	2011	USA	C	284	CPP	<50	LPS (biopsy or visual proven), C, patients’ symptoms, physical examination, PUF scores	Menopausal status, attempt to conceive, non-gynecological or non-lower urinary tract problems, previous bilateral oophorectomy	172	-
Rackow [40]	2009	USA	C	28	CPP	13–25	LPS (visual proven), C, BH	-	7	-
Stanford [41]	2005	USA	C	64	CPP	32	C, BH, LPS (biopsy proven), PST	-	27	12
Smorgick [42]	2013	USA	C	138	Endometriosis	<21	LPS (biopsy or visual proven), medical records	Unclear diagnosis, incomplete records, endometriosis caused by obstructive Mullerian anomalies	11	-
Warren [43]	2009	USA	C	313	IC/PBS and controls	18–76	Medical and surgical records	More than 12 months of IC/PBS symptoms	62	6
Wu [44]	2018	TW	C	36,764	Endometriosis and controls	18–50	Medical records	Positive urinary cultures	18	18

Study design is described as cohort study (C), including also prospective and retrospective studies. Follow-up (FUP) is expressed in months. Chronic pelvic pain (CPP). Cases = patients with coexistence between endometriosis and IC/PBS. Cystoscopy (C) with bladder hydrodistention (BH), laparoscopy (LPS), potassium sensitivity test (PST), pelvic pain and urgency/frequency patient symptom scale (PUF). AU = Australia, TW = Taiwan. NA = not available, the diagnosis was evaluated through medical records. Age is expressed in years old, mean or range values are reported. Follow-up is expressed in months.

**Table 3 healthcare-12-02403-t003:** Studies evaluating the prevalence of the coexistence of endometriosis and IC/PBS in patients with CPP.

Authors	N	Prevalence of Endometriosis	Prevalence of IC/PBS	Prevalence of the Coexistence	Overlapping Symptoms	Proposed Treatment
Cheng [31]	150	60% (90/150)	53% * (80/150)	30% (45/150)	CPP, urinary symptoms	-
Chung [32]	178	75% (134/178)	89.3% (159/178)	64.6% (115/178)	CPP, bladder base and uterine tenderness, with or without urinary symptoms	-
Chung [33]	60	93.3% (56/60)	96.6% (58/60)	78.3% (47/60)	dyspareunia, dysmenorrhea with or without urinary symptoms	-
Clemons [34]	45	46.6% (21/45)	37.8% (17/45)	15.5% (7/45)	dyspareunia, dysmenorrhea, urgency, frequency, nicturia	-
Paulson [38]	162	76% (123/162)	82% (133/162)	66% (107/162)	CPP	Medical or surgical treatment of endometriosis, cystoscopy for IC/PBS and if necessary, anticholinergic medications, antihistamines, intravesical instillations
Paulson [39]	284	78% (216/284)	81% (225/284)	60.5% (172/284)	CPP, anterior vaginal wall tenderness (AVWT)	-
Rackow [40]	28	64.2% (18/28)	39.2% (11/28)	25% (7/28)	CPP, dysmenorrhea, dyspareunia, nocturia, urgency, dysuria	-
Stanford [41]	64	73.4% ** (47/64)	68.7% (44/64)	42% ** (27/64)	CPP	-

Prevalence is expressed as %, (n). * according to ESSIC criteria ** evaluated in patients both with endometriosis or adhesions.

## Data Availability

Analyzed or generated during the study and can be requested from the corresponding author upon reasonable request.

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
