# Peer review of "The Evil Twins of Chronic Pelvic Pain Syndrome: A Systematic Review and Meta-Analysis on Interstitial Cystitis/Painful Bladder Syndrome and Endometriosis"

_healthcare, 2024, doi:10.3390/healthcare12232403_

Round 1
Reviewer 1 Report
Comments and Suggestions for Authors
Dear Authors
Thanks for your Practical review. chronic pelvic is one of the most challenging situation for gynecologist.There are some points need to be considered.
1-It is better to clarify inclusion and exclusion criteria in method part.
2-please include more updated references in your study. There are many Studies published before 2016.
3-It is better to mention limitation and strengths of study in discussion part. 4-please adress the bias of primary publications.
5-According to this review what is your recommendation for early diagnosis and prevention of Chronic Pelvic Pain Syndrome
Author Response
COMMENT 1-It is better to clarify inclusion and exclusion criteria in method part.
RESPONSE 1 - Thank you very much for your observation, we put the inclusion and exclusion criteria in the method part of the manuscript.
COMMENT 2-please include more updated references in your study. There are many Studies published before 2016.
RESPONSE 2 - We added some relevant studies published before 2016 to the discussion section.
COMMENT 3-It is better to mention limitation and strengths of study in discussion part.
RESPONSE 3- We thank the reviewer for this observation, we changed the manuscript.
COMMENT 4-please adress the bias of primary publications.
RESPONSE 4 - As we stated in the manuscript, the main biases are the different inclusion and exclusion criteria, the fact that the diagnostic criteria are not the same for all the studies, and the differences in the referral centres between studies.
COMMENT 5-According to this review what is your recommendation for early diagnosis and prevention of Chronic Pelvic Pain Syndrome
RESPONSE 5- We suggest that even if a formal diagnosis is made, women always need to be screened for the possible coexistence of other diseases, above all in women who are unresponsive to treatments or partially responsive. This is particularly important for women with endometriosis, for whom the final stage is demolition surgery, which, in the case of the coexistence of IC/PBS, may be unnecessary.
Reviewer 2 Report
Comments and Suggestions for Authors
Dear authors, it is a very interesting and comprehensive review. However, what is the main question of your review? Knowing that the main diagnostic tool for endometriosis is histopathology after laparoscopy, do you see an urgent need for this in case of CPP? Also, as you emphasized, in some studies they included patients with CPP and bladder base/anterior vaginal wall and uterine tenderness with or without voiding symptoms,endometriosis and IC/PBS in this study could be overestimated.
You say the coexistence between endometriosis and IC/PBS is frequent regardless of the population studied. For this reason, over the years, it is possible that many useless surgeries have been performed in patients with CPP for the differential diagnosis.What kind of useless surgery?
In addition, in this study women with IC/PBS were more commonly diagnosed with endometriosis and fibroids than controls.As you see also other conditions are linked to CPP.How do you comment?
The results of the meta-analysis show that the prevalence of IC/PBS in women with endometriosis is higher than in women without endometriosis. However, the result lacks statistical significance (OR 0,82; IC 0,54-1,26). This may be linked to the heterogeneity of the prevalence of the studies included in the meta-analysis . According to these findings what is the relevance of this metaanaysis?
The study of Rackow et al. [39] shows a prevalence of 30%. However, the main difference in this study is that they included only young women under 25 years old . This is an important bias.
As you acknowledged, first-line treatment of IC/PBS is with oral drugs (pentosan polysulfate sodium, hydroxyzine, amitriptyline, pregabalin). Several intravesical therapies can be used, such as intravesical hyaluronic acid/chondroitin sulfate, in combination or not with oral drugs [58] In this situation, a useless laparoscopy to find a possible endometriosis is still a need?
The main limitation is the variability of the inclusion criteria used in different studies and the bias linked to probably the kind of referral (urogynecological centre, pelvic pain/endometriosis centre).This is a good observation.
Thank you.
Author Response
COMMENT 1 - Dear authors, it is a very interesting and comprehensive review. However, what is the main question of your review? Knowing that the main diagnostic tool for endometriosis is histopathology after laparoscopy, do you see an urgent need for this in case of CPP?
RESPONSE 1 - We do not believe that all women with chronic pelvic pain (CPP) should undergo a diagnostic laparoscopy, as suggested by the new ESHRE guidelines. It is important to emphasize that even after a diagnosis is made—whether it is interstitial cystitis/bladder pain syndrome (IC/PBS) or endometriosis—gynecologists should be aware of the high prevalence of coexistence between these conditions. This overlap in symptoms can lead to diagnostic delays, which we have analyzed in many studies included in this review.
COMMENT 2- Also, as you emphasized, in some studies they included patients with CPP and bladder base/anterior vaginal wall and uterine tenderness with or without voiding symptoms,endometriosis and IC/PBS in this study could be overestimated.
RESPONSE 2- We thank the reviewer for raising this point, in the manuscript it is amended that this is a selection bias. However, it is not much different from clinical practice, since even 39% of women with only endometriosis (and not IC) have anterior vaginal wall tenderness (AVWT).
COMMENT 3- You say the coexistence between endometriosis and IC/PBS is frequent regardless of the population studied. For this reason, over the years, it is possible that many useless surgeries have been performed in patients with CPP for the differential diagnosis. What kind of useless surgery?
RESPONSE 3- We thank the reviewer for raising this point; we can assume that physicians all over the world propose surgery for the diagnosis and consequent treatment of endometriosis unresponsive to treatments in women with CPP. We amended this point in the manuscript.
COMMENT 4- In addition, in this study women with IC/PBS were more commonly diagnosed with endometriosis and fibroids than controls. As you see also other conditions are linked to CPP. How do you comment?
RESPONSE 4-This is an excellent observation. However, it is rare that a woman with fibroids goes into a diagnostic delay of her condition; therefore, even if there is a coexistence, the presence of fibroids is documented in most cases. On the other side, endometriosis and IC/PBS both recognize a diagnostic delay in their diagnosis, and this has a different clinical impact on women’s lives.
COMMENT 5- The results of the meta-analysis show that the prevalence of IC/PBS in women with endometriosis is higher than in women without endometriosis. However, the result lacks statistical significance (OR 0,82; IC 0,54-1,26). This may be linked to the heterogeneity of the prevalence of the studies included in the meta-analysis . According to these findings what is the relevance of this metaanaysis?
RESPONSE 5-This meta-analysis strongly encourages the multidisciplinary approach to women with CPP to prevent the central sensitization, the “chronicity” and the detrimental effects on the life of women. It focuses on the fact that even if a cause of CPP is found, CPP needs to be considered more as a systemic disease with many aspects and many symptoms which need different treatments. The fact that the results suggest that it is more common to find the coexistence of these two conditions instead of one clinical disease alone, marks the point that we should not stop at the “first” diagnosis.
COMMENT 6-The study of Rackow et al. [39] shows a prevalence of 30%. However, the main difference in this study is that they included only young women under 25 years old . This is an important bias.
RESPONSE 6-We thank the reviewer for this observation; we address this critical bias in the manuscript. However, in many studies, the age range included women under 25.
COMMENT 7-As you acknowledged, first-line treatment of IC/PBS is with oral drugs (pentosan polysulfate sodium, hydroxyzine, amitriptyline, pregabalin). Several intravesical therapies can be used, such as intravesical hyaluronic acid/chondroitin sulfate, in combination or not with oral drugs [58] In this situation, a useless laparoscopy to find a possible endometriosis is still a need?
RESPONSE 7-We do not believe that all women should undergo a diagnostic laparoscopy for CPP, as the new ESHRE guidelines suggest. Indeed, we want to emphasize that even if a diagnosis is made for CPP, it is not improbable that other pathologies are coexistent, especially if the patient is unresponsive or only partially responsive to therapies.
Reviewer 3 Report
Comments and Suggestions for Authors
The paper is well-organized, following the PRISM criteria recommendations. The studies included in the paper are relevant and of good quality. The introduction is excellent, as is the methodology. The results are clearly presented. The main critique lies in the discussion, where the authors cited numerous relevant studies, explaining the overlap in prevalence and clinical presentation of these two different entities, but treatment was not mentioned. It was only briefly referenced in one sentence, stating that hormonal preparations are used for individuals with endometriosis and that the duration of symptoms is 9.5 years. It would be beneficial if, in addition to the tabular presentation of incidence (Table 3), they could also present the overlapping symptoms of these two entities and the treatment methods in a similar format.
Author Response
COMMENT 1- The paper is well-organized, following the PRISM criteria recommendations. The studies included in the paper are relevant and of good quality. The introduction is excellent, as is the methodology. The results are clearly presented. The main critique lies in the discussion, where the authors cited numerous relevant studies, explaining the overlap in prevalence and clinical presentation of these two different entities, but treatment was not mentioned. It was only briefly referenced in one sentence, stating that hormonal preparations are used for individuals with endometriosis and that the duration of symptoms is 9.5 years. It would be beneficial if, in addition to the tabular presentation of incidence (Table 3), they could also present the overlapping symptoms of these two entities and the treatment methods in a similar format.
RESPONSE 1-We thank the reviewer for raising this point, we provided the information to the table. Unfortunately, just one study of the list included also a statement about the treatment of IC/PBS and endometriosis.
Reviewer 4 Report
Comments and Suggestions for Authors
Thanks for the opportunity to review this manuscript!
The topic of chronic pelvic pain is complex and worthy of more attention in the literature. I commend the authors for their work in this field.
Although the objective of estimation of the comorbid prevalence of IC/PBS and endometriosis is potentially helpful, the literature is so markedly heterogeneous that I think it is a question that is very difficult to answer via meta-analysis. If you are going to try to answer this question, you need to more clearly define in your methodology what your inclusion and exclusion criteria are, what types of studies (more detail than just 'prospective or retrospective') you will consider, and how you will address the marked heterogeneity that you anticipate you will encounter, as well as the confounders/effect modifiers that may affect the relationship between endometriosis and IC/PBS.
You have also excluded almost half of potentially inclusion worthy studies because they are not english language- did you try to assess/appreciate what information is in these studies? might it be worth translating some if they look to be of potentially high quality?
Looking at your forest plot, I think you have presented the information backwards. When your odds ratios considering odds of PBS in women with endo vs. odds of PBS in women without endo come out below 1, it is suggesting that there are higher odds of PBS in women WITHOUT endo, which doesn't come across in the forest plot as laid out... did you label it backwards?

There are a number of grammatical and spelling errors that make the manuscript somewhat awkward to read.
Author Response
COMMENT 1- Although the objective of estimation of the comorbid prevalence of IC/PBS and endometriosis is potentially helpful, the literature is so markedly heterogeneous that I think it is a question that is very difficult to answer via meta-analysis. If you are going to try to answer this question, you need to more clearly define in your methodology what your inclusion and exclusion criteria are, what types of studies (more detail than just 'prospective or retrospective') you will consider, and how you will address the marked heterogeneity that you anticipate you will encounter, as well as the confounders/effect modifiers that may affect the relationship between endometriosis and IC/PBS.
RESPONSE 1- We are aware of the limitations and bias of this meta-analysis. However, all studies go into the same direction, suggesting that this coexistence is usually unpredictable and under diagnosed. We added a statement in the “Materials and methods” section about inclusion and exclusion criteria. The heterogeneity and the bias we encountered in our meta-analysis are already explained in the “Discussion” section.
COMMENT 2- You have also excluded almost half of potentially inclusion worthy studies because they are not english language- did you try to assess/appreciate what information is in these studies? might it be worth translating some if they look to be of potentially high quality?
RESPONSE 2- Thank you for this suggestion. Most of the studies we excluded for language issues were reviews. We believe that most original studies are published in English to reach a wider international audience.
COMMENT 3-Looking at your forest plot, I think you have presented the information backwards. When your odds ratios considering odds of PBS in women with endo vs. odds of PBS in women without endo come out below 1, it is suggesting that there are higher odds of PBS in women WITHOUT endo, which doesn't come across in the forest plot as laid out... did you label it backwards?
RESPONSE 3- We thank you the reviewer for you suggestion, we adjusted the meta-analysis.
RESPONSE 4- We made the changes to the manuscript according to the pdf submitted.
Reviewer 5 Report
Comments and Suggestions for Authors
Thank you for your paper. I have some comments:
-This topic is not novel. Please mention the novel point that this paper should be more valuable than other papers. Tirlapur SA, Kuhrt K, Chaliha C, Ball E, Meads C, Khan KS. The 'evil twin syndrome' in chronic pelvic pain: a systematic review of prevalence studies of bladder pain syndrome and endometriosis. Int J Surg. 2013;11(3):233-237. doi:10.1016/j.ijsu.2013.02.003
-In introduction, please also mention the pelvic pain in other gynecologic diseases such as intracavitary pathologies. Nguyen PN, Nguyen VT. Evaluating Clinical Features in Intracavitary Uterine Pathologies among Vietnamese Women Presenting with Peri-and Postmenopausal Bleeding: A Bicentric Observational Descriptive Analysis. J Midlife Health. 2022 Jul-Sep;13(3):225-232. doi: 10.4103/jmh.jmh_81_22. Epub 2023 Jan 14. PMID: 36950211; PMCID: PMC10025815.
-In Table 2, please reconsider this statement. “Study design is described as prospective (P), retrospective (R) or cohort study (C).” Prospective/retrospective/cross-sectional study could be combined with cohort study.
-In Figure 2, please rearrange the studies following the order of published year, similar to table 3.
-Please place the strengths and limitations at the end of discussion.
-Please insert the reference citation following journal’s style.
Author Response
COMMENT 1- This topic is not novel. Please mention the novel point that this paper should be more valuable than other papers. Tirlapur SA, Kuhrt K, Chaliha C, Ball E, Meads C, Khan KS. The 'evil twin syndrome' in chronic pelvic pain: a systematic review of prevalence studies of bladder pain syndrome and endometriosis. Int J Surg. 2013;11(3):233-237. doi:10.1016/j.ijsu.2013.02.003
RESPONSE 1- We thank the reviewer for raising this issue. The manuscript by Tirlapur et al. was of great quality and interest. Indeed, it focused on the fact that PBS is more common in CPP patients than what was thought before. More than 10 years after its publication, the impact of IC/PBS in the development of CPP is well known. However, our work points out the need for more studies since after 2013 (when the study of Tilarpur et al. was published), just two other studies were made on this topic, despite being one of the most relevant topics in gynaecology on which a lot is still unknown. We added a statement about this in our manuscript.
COMMENT 2- In introduction, please also mention the pelvic pain in other gynecologic diseases such as intracavitary pathologies. Nguyen PN, Nguyen VT. Evaluating Clinical Features in Intracavitary Uterine Pathologies among Vietnamese Women Presenting with Peri-and Postmenopausal Bleeding: A Bicentric Observational Descriptive Analysis. J Midlife Health. 2022 Jul-Sep;13(3):225-232. doi: 10.4103/jmh.jmh_81_22. Epub 2023 Jan 14. PMID: 36950211; PMCID: PMC10025815.
RESPONSE 2-Thank you very much for your suggestion, we added a statement and the subsequent citation.
COMMENT 3- In Table 2, please reconsider this statement. “Study design is described as prospective (P), retrospective (R) or cohort study (C).” Prospective/retrospective/cross-sectional study could be combined with cohort study.
RESPONSE 3- We changed the label of the study design according to this request.
COMMENT 4-In Figure 2, please rearrange the studies following the order of published year, similar to table 3.
RESPONSE 4-We rearranged the studies following the order of published year.
COMMENT 5-Please place the strengths and limitations at the end of discussion.
RESPONSE 5- We placed the strengths and limitations where requested.
COMMENT 6-Please insert the reference citation following journal’s style.
RESPONSE 6-We adjusted the reference citation style.
Round 2
Reviewer 4 Report
Comments and Suggestions for Authors
Thanks for the opportunity to review this revised manuscript
I really agree with the value of discussing the comorbidity of endometriosis and IC/PBS, but am still struggling to follow the meta-analytic component of this manuscript.
Where in the body of the manuscript is there a meta-analytic calculation that generates the average coexistence of IC/PBS and endometriosis at 51.25 +/- 22.73%?
Similarly, I still cannot understand Figure 2 the forest plot. For example, if in Clemons 2002 the events (PBS/IC) in patients without endometriosis were 8 of a total 14 patients, and events in those WITH endometriosis were 7 of 21 patients, how can your odds ratio come out indicating that the odds of IC/PBS are higher in patients with endometriosis? This is backwards... the studies you included in Figure 2 actually show that the odds of PBS/IC are lower in patients with endometriosis than in those without endometriosis (perhaps not statistically significant, but that is the trend unless I am really missing something).
Also, broadly, conclusions do not follow from results (see abstract conclusion "Women with CPP need a multidsiciplinary approach and a referral to centers with specific expertise. In cases of endometriosis unresponsive to treatement, other reasons for CPP need to be ruled out." These are major reaches from the finding that 15 to 78 % of women with CPP have coexistant endometriosis and IC/PBS- maybe comment more factually that there is highly heterogeneous data on prevalence, but that the coexistence of these two pain conditions should be considered or something like that?
Looking at this manuscript again as well, I don't see any formal assessment of risk of bias of the included studies which would be a typically incldued component of a systematic review/metanalysis framework.
I wonder if you could significantly strengthen your manuscript by simply dropping the meta-analytic components of the study with the justification that the heterogeneity of the studies was simply too great, and instead present a more descriptive summary of your systematic review results +/- present this as more of a narrative review vs. systematic review/meta analysis. Or, perhaps work with a statistician to clarify your calculations and presentation of data?
Author Response
Where in the body of the manuscript is there a meta-analytic calculation that generates the average coexistence of IC/PBS and endometriosis at 51.25 +/- 22.73%?
Thank you, we reviewed with the statistician and removed this data from the abstract
Similarly, I still cannot understand Figure 2 the forest plot. For example, if in Clemons 2002 the events (PBS/IC) in patients without endometriosis were 8 of a total 14 patients, and events in those WITH endometriosis were 7 of 21 patients, how can your odds ratio come out indicating that the odds of IC/PBS are higher in patients with endometriosis? This is backwards... the studies you included in Figure 2 actually show that the odds of PBS/IC are lower in patients with endometriosis than in those without endometriosis (perhaps not statistically significant, but that is the trend unless I am really missing something).
We thank the reviewer for this observation. We reviewed with the statistician the meta-analysis. There was a typos error in the legend of the forest plot which we corrected.
Also, broadly, conclusions do not follow from results (see abstract conclusion "Women with CPP need a multidsiciplinary approach and a referral to centers with specific expertise. In cases of endometriosis unresponsive to treatement, other reasons for CPP need to be ruled out." These are major reaches from the finding that 15 to 78 % of women with CPP have coexistant endometriosis and IC/PBS- maybe comment more factually that there is highly heterogeneous data on prevalence, but that the coexistence of these two pain conditions should be considered or something like that?
We changed the conclusion of the abstract according to this suggestion “Prevalence data about coexistence of endometriosis and IC/PBS are highly heterogeneous, probably due to the paucity of available data. However, in cases of endometriosis unresponsive to treatment, other reasons for CPP (such as IC/PBS) need to be ruled out.”
Looking at this manuscript again as well, I don't see any formal assessment of risk of bias of the included studies which would be a typically incldued component of a systematic review/metanalysis framework.
We think that in consideration of the limited number of studies included and heterogenity of data prevalence, a formal bias risk analysis is not likely adding much to the manuscript.
I wonder if you could significantly strengthen your manuscript by simply dropping the meta-analytic components of the study with the justification that the heterogeneity of the studies was simply too great, and instead present a more descriptive summary of your systematic review results +/- present this as more of a narrative review vs. systematic review/meta analysis. Or, perhaps work with a statistician to clarify your calculations and presentation of data?
We do think that after statistician reanalysis, meta-analysis could still be a precious add to the manuscript.